



# Mass spectrometry-based *aerosolomics*: a new approach to resolve sources, composition, and partitioning of secondary organic aerosol

Markus Thoma[1], Franziska Bachmeier[1], Felix Leonard Gottwald[1], Mario Simon[1], and Alexander Lucas Vogel[1]

[1]Institute for Atmospheric and Environmental Sciences, Goethe-University Frankfurt, 60438 Frankfurt am Main, Germany

**Correspondence:** Alexander L. Vogel (vogel@iau.uni-frankfurt.de)

**Abstract.** Particulate matter (PM) largely consists of secondary organic aerosol (SOA) that is formed via oxidation of biogenic and anthropogenic volatile organic compounds (VOCs). Unambiguous identification of SOA molecules and their assignment to their precursor vapors is a challenge that has so far only succeeded for a few SOA marker compounds, which are now well characterized and (partly) available as authentic standards. In this work, we resolve the complex composition of SOA

by a top-down approach based on a newly created *aerosolomics* database, which is fed by non-target analysis results of filter samples from oxidation flow reactor experiments. We investigated the oxidation products from the five biogenic VOCs α-pinene, β-pinene, limonene, 3-carene, and *trans*-caryophyllene and from the four anthropogenic VOCs toluene, *o*-xylene, 1,2,4-trimethylbenzene, and naphthalene. Using ultra-high performance liquid chromatography coupled to a high-resolution (Orbitrap) mass spectrometer, we determine the molecular formula of 596 chromatographically separated compounds based

on exact mass and isotopic pattern. We utilize retention time and fragmentation mass spectra as a basis for unambiguous attribution of the oxidation products to their parent VOCs. Based on the molecular-resolved application of the database, we are able to assign roughly half of the total signal of oxygenated hydrocarbons in ambient suburban $PM_{2.5}$ to one of the nine studied VOCs. The application of the database enabled us to interpret the appearance of diurnal compound clusters that are formed by different oxidation processes. Furthermore, the application of a hierarchical cluster analysis (HCA) on the same set

of filter samples enabled us to identify compound clusters that depend on sulfur dioxide mixing ratio and temperature. This study demonstrates how *aerosolomics* tools (database and HCA) applied on PM filter samples can improve our understanding of SOA sources, their formation pathways, and temperature-driven partitioning of SOA compounds.

## 1 Introduction

Secondary organic aerosol (SOA) is a complex mixture forming through the oxidation of biogenic (BVOCs) and anthropogenic

volatile organic compounds (AVOCs) in the atmosphere. Aerosol particles influence Earth's climate as well as human health (Hallquist et al., 2009; Shrivastava et al., 2017). Earlier work has shown that SOA makes up a large fraction of fine particulate matter ($PM_{2.5}$, particles with an aerodynamic diameter less than $2.5\,\mu m$) (Jimenez et al., 2009; Huang et al., 2014; McDonald et al., 2018). Globally, the emissions of BVOCs are considerably higher than those of AVOCs, with $760\text{--}1150\,TgC\,y^{-1}$ compared to $140\,TgC\,y^{-1}$, respectively (Kari et al., 2019; Shrivastava et al., 2017; Sindelarova et al., 2014). BVOC emissions





can mainly be distributed among isoprene (70 %), monoterpenes (11 %), methanol (6 %), and others (13 %) (Sindelarova et al.,
2014). AVOCs and BVOCs are not only emitted from different sources, but they also have different SOA yields, and result in
different products with distinct different properties. Furthermore, it is known that both organic and inorganic anthropogenic
emissions can affect SOA formation from BVOCs (Kari et al., 2019; Xu et al., 2021). Still major knowledge gaps exist on the
sources and formation pathways of SOA, its transformation and lifetime in the atmosphere, and its underlying effect on Earth's

climate and human health. Furthermore, emissions of anthropogenic and biogenic precursors as well as their atmospheric fate
are uncertain, resulting in a discrepancy between measured and modeled SOA (Fuzzi et al., 2015). Improved chemical char-
acterization of ambient SOA can help understanding of sources, formation pathways, and effects on both climate and human
health.

        Many controlled laboratory studies have increased our mechanistic understanding of the oxidation of volatile organic com-

pounds (VOCs) (Burkholder et al., 2017). However, the ambient atmosphere is usually more complex than chamber experi-
ments, and unaccounted chemical interactions can therefore alter SOA yields that are derived from simple laboratory systems
(McFiggans et al., 2019). Furthermore, it is likely that many VOCs are understudied that are relevant for SOA formation.
Therefore, a comprehensive top-down investigation of SOA can enable identification of missing important precursor gases or
relevant formation pathways. Numerous previous investigations (Glasius et al., 2000; Kristensen et al., 2016; Nozière et al.,

2015; Surratt et al., 2007) clearly highlight the advantages of offline measurement techniques, which apply separation tech-
niques like gas chromatography or (ultra-high performance) liquid chromatography (UHPLC) coupled to (high-resolution)
mass spectrometry (HRMS), because with these techniques the unambiguous identification of different compounds becomes
possible. In recent years, non-target analysis (NTA) of UHPLC-HRMS measurements has become a powerful tool that builds
peak lists of all detected compounds in complex samples, and determines the molecular formula based on the exact mass and

isotopic pattern. Furthermore, MS[2]-spectra can be compared to fragmentation libraries and enable database-assisted identifi-
cation of compounds (Ditto et al., 2018; Ma et al., 2022; Pereira et al., 2021; Pleil et al., 2018). However, there are currently
no established databases of atmospheric SOA tracers, which can be applied on measurements of ambient $PM_{2.5}$ filter samples.

        Therefore, we initialized a database for compound matching, based on filters from potential aerosol mass (PAM) oxida-
tion flow reactor (OFR) experiments of nine biogenic and anthropogenic VOCs. We applied the database to ambient air filter

samples collected in summer 2018 near Vienna (Austria). Figure 1 shows the principal steps of the new *aerosolomics* ap-
proach that is based on the comparison between filter samples from OFR experiments and from the ambient. Additionally,
a hierarchical cluster analysis (HCA) was performed in order to reduce the complexity of the ambient dataset and to assign
compounds to certain formation processes or emission sources. Both strategies combined allow the identification of oxidation
products from either biogenic or anthropogenic VOCs, and enable a better understanding of the oxidation conditions and of

temperature-driven gas-to-particle partitioning.



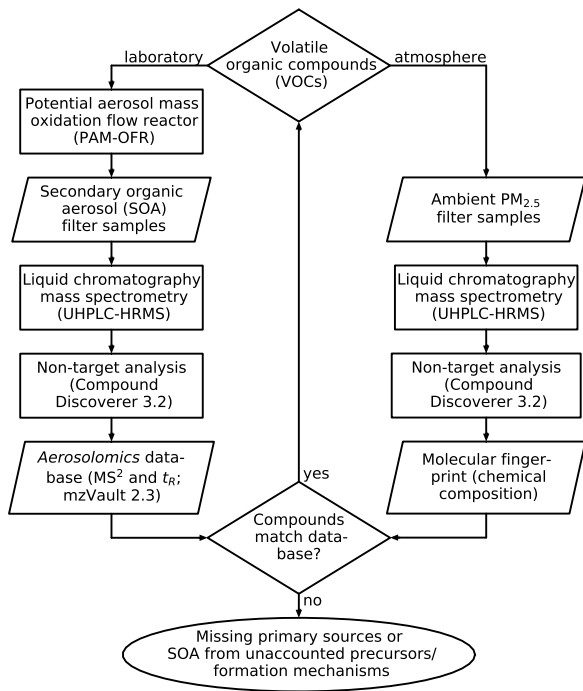

**Figure 1.** Establishment and application of the *aerosolomics* database: The database is filled with the results of several PAM-OFR experiments with different precursors and reactants and compared to the molecular fingerprints of ambient $PM_{2.5}$ filter samples. Matching compounds can be assigned to the corresponding VOC precursor. Knowledge about not-matching compounds, due to missing primary sources or SOA from unaccounted precursors/formation mechanisms, can guide further experiments.

## 2 Experimental

### 2.1 Oxidation flow reactor experiments

We used a PAM-OFR (Aerodyne Research Inc.) (Kang et al., 2007; Lambe et al., 2011) for laboratory SOA formation of biogenic VOCs (-)-$\alpha$-pinene (98 %, Alfa Aesar, CAS: 7785-26-4), (+)-$\beta$-pinene (98 %, Sigma-Aldrich, CAS: 19902-08-0), (S)-(-)-limonene (96 %, Sigma-Aldrich, CAS: 5989-54-8), (+)-3-carene ($\geq$ 98.5 %, Fluka Analytical, CAS: 498-15-7), and (-)-*trans*-caryophyllene ($\geq$ 98 %, Sigma-Aldrich, CAS: 87-44-5), as well as the anthropogenic VOCs toluene (> 99.5 %, Fluka Analytical, CAS: 108-88-3), *o*-xylene (> 99.0 %, Fluka Analytical, CAS: 95-47-6), 1,2,4-trimethylbenzene (98 %, Sigma-Aldrich, CAS: 95-63-6), and naphthalene (> 99 %, Merck KGaA, CAS: 91-20-3). In dark ozonolysis experiments, all BVOCs were oxidized with externally produced ozone ($O_3$). Hydroxyl radicals (OH) formed during ozonolysis of VOCs were not scavenged. Under daytime simulations, OH was generated by UV light ($\lambda = 254\,\mathrm{nm}$) inside the OFR.

We evaporated the VOCs in a heated glass flask purged continuously with nitrogen ($N_2$, 6.0 purity, Nippon Gases). The aerosol mass concentration was measured with a scanning mobility particle sizer spectrometer (SMPS; consisting of an elec-





trostatic classifier 3082, a differential mobility analyzer 3081A, and an ultrafine condensation particle counter 3776, TSI Inc.).
Changing the precursor concentration by varying the temperature inside the flask resulted in aerosol mass concentrations be-
tween 20 and $184\,\mu g\,m^{-3}$. The individual settings and the resulting mass concentrations are given in Table S1.

In all experiments the nitrogen flow into the reactor was $4.8\,L\,min^{-1}$, the oxygen ($O_2$, 5.0 purity, Nippon Gases) flow
was $1.2\,L\,min^{-1}$ resulting in a mean residence time of 2.4 minutes. The relative humidity was $55\,\%$. The $O_3$ concentration
was $\sim$1 ppm, decreasing to 0.8 ppm under OH conditions. As an approximation of the mean OH exposure, we carried out a
calibration experiment, measuring the decay of $SO_2$ at different irradiances in the range between 0 and $200\,\mu W\,cm^{-2}$, adapted
to Li et al. (2015). The calculated OH exposures and the fitted curve is shown in Fig. S1. The experimental settings resulted in
a mean OH exposure of $1.06 \times 10^{12}$ molecules $cm^{-3}\,s^{-1}$, corresponding to approximately 11 days of equivalent atmospheric
OH exposure, based on the assumption of an averaged tropospheric OH concentration of $1.09 \times 10^{6}$ molecules $cm^{-3}$ (Li et al.,
2018). The aged air leaving the OFR passed through two $50\,cm$ denuders, packed with charcoal (IAC-402, Infiltec GmbH) and
potassium permanganate ($KMnO_4$, IAC-630, Infiltec GmbH) in order to remove (reactive) gas-phase compounds. Glass fibre
filters ($47\,mm$, Pallflex Emfab Filters, Pall) sampled the formed SOA particles with a flow of $3\,L\,min^{-1}$ for a duration of
minutes. Until sample preparation and analysis, filter samples were packed in aluminium foil and stored at $-18\,°C$.

## 2.2 Ambient air filter sampling campaign

$PM_{2.5}$ filter samples were collected in August 2018 during a field campaign ($48.127°$ N, $16.534°$ E), at an suburban background
station between the Vienna International Airport in the east and the Schwechat Industrial Park and Vienna city center in
the north-west. 52 glass fiber filters ($150\,mm$, Ahlstrom-Munksjö) were sampled for 12 hours starting at 05:00 (UTC) or
17:00 (UTC) respectively, using a high volume sampler (DHA-80, Digitel Elektronik AG) at a flow rate of $30\,m^3\,h^{-1}$. The
meteorological parameters (i.e., wind direction, wind speed, and temperature), the trace gas concentration (i.e., nitrogen oxide
(NO), nitrogen dioxide ($NO_2$), sulfur dioxide ($SO_2$), and carbon monoxide (CO)), as well as the $PM_{2.5}$ mass concentration
were monitored continuously.

## 2.3 Sample preparation

From each ambient filter sample, one punch ($25\,mm$ diameter) was cut in small pieces and extracted in a glass vial using $200\,\mu L$
of ultrapure water (Milli-Q Reference A+, Merck KGaA) and methanol (Optima LC/MS Grade, Thermo Fisher Scientific Inc.)
(90/10, v/v) for 20 minutes on an orbital shaker with $300\,rpm$. Afterwards, the solvent was drawn up with a syringe (Injekt-F,
Braun Melsungen AG) and filtered through a $0.2\,\mu m$ syringe filter (non-sterile PTFE Syringe Filter, Thermo Fisher Scientific
Inc.). In a second step, $100\,\mu L$ of the solvent mixture was added and the procedure was repeated. $50\,\mu L$ of the extracted sample
was mixed with $5\,\mu L$ of an internal standard containing isotopically labeled benzoic acid ($C_6H_5{}^{13}CO_2H$, 99 atom $\%$ $^{13}C$,
Sigma-Aldrich, $c = 0.1\,mg\,mL^{-1}$).

Half of each filter from the OFR experiments was cut in small pieces and extracted analogously to the ambient air filter
samples with an adjustment in the eluent volume: $180\,\mu L$ in the first and $80\,\mu L$ in the second step was used. Finally $100\,\mu L$ of
the extracted sample was mixed with $10\,\mu L$ of the internal standard.


## 2.4 Standard mixture for non-target analysis validation

A solution of 13 analytical standards was used to validate UHPLC-HRMS measurements and the NTA workflow with primary
attention toward automated compound identification, but also toward fragmentation and adduct formation, which can result in
false positives. To cover a variety of atmospherically relevant compounds, the mixture consists of carboxylic acids, organosul-
fates and -phosphates, as well as nitrogen containing compounds. The injection volume for the analysis was $1\,\mu L$. A detailed
overview of the substances used and their concentrations in the mixture are given in Table S2.

## 2.5 UHPLC-($-$)HESI-HRMS measurements

The extracts of the ambient PM samples were separated by ultra-high performance liquid chromatography (Vanquish Flex,
Thermo Fisher Scientific Inc.) on a reversed phase column (Accucore $C_{18}$, $2.6\,\mu m$, $150 \times 2.1\,mm$, Thermo Fisher Scientific
Inc.), ionized in the negative mode using a heated electrospray ionization source (HESI-II Probe, Thermo Fisher Scientific
Inc.) and detected with a high-resolution hybrid quadrupole-Orbitrap mass spectrometer (Q Exactive Focus, Thermo Fisher
Scientific Inc.). Eluents were ultrapure water (eluent A) and methanol (eluent B), both mixed with $0.1\,\%$ formic acid ($98\,\%$,
Merck KGaA). The injection volume was $5\,\mu L$, the flowrate was $400\,\mu L\,min^{-1}$, and the temperature was $40\,°C$. The gradient
started with $1\,\%$ eluent B (0–0.5 min), increased linearly to $99\,\%$ B (0.5–14 min), stayed at $99\,\%$ B (14–16 min), backflushed in
one minute and equilibrated in three minutes resulting in a total run time of 20 minutes. The ion source settings were $-3.5\,kV$
spray voltage, $40\,psi$ sheath gas, 8 arbitrary units auxiliary gas, and $262.5\,°C$ capillary temperature. The spectra were recorded
in full scan MS with data dependent tandem mass spectrometry (ddMS$^2$) using a higher-energy collisional dissociation (HCD)
cell with stepped collision energies of 15, 30 and $45\,eV$. The scan range in full MS was $m/z$ 50–750 with a resolution of $70\,000$
at $m/z$ 200. For ddMS$^2$ the resolution was $17\,500$.

A representative selection of 10 ambient PM samples was measured a second time. The selection was based on external
influences like wind direction, temperature, time of day and trace gas concentrations. In one sequence, together with filter
samples from laboratory OFR experiments, we applied and improved gradient on another reversed phase column (Cortecs
Solid Core T3, $2.7\,\mu m$, $150 \times 3\,mm$, with the corresponding VanGuard Cartridge, Waters Corp.). The gradient also started
with $1\,\%$ B for half a minute, increased linearly to $99\,\%$ B in 15 minutes, and held it for 2 minutes. Afterwards, the column was
backflushed in $90\,seconds$ and equilibrated in two and a half minutes resulting in a total run time of 21.5 minutes. This dataset
is basis for the application of the database on ambient samples.

As a quality control routine, we extracted one filter three times to estimate the reproducibility of the extraction procedure.
In addition, we determined the instrument performance by a triplicate measurement of one filter extract. We calculated the
relative standard deviation (RSD) for 7 compounds ($m/z$ 115–357 and signal intensities of $3 \times 10^5$–$5 \times 10^8$ counts). Averaged
over all seven compounds, we determined a mean RSD of $6.7\,\%$ for the reproducibility of the extraction procedure and $2.1\,\%$
for the instrumental performance.





## 2.6 Non-target analysis, MS$^2$ libraries, hierarchical cluster analysis, and volatility estimation

We used Compound Discoverer 3.2 (Thermo Fisher Scientific Inc.) for the NTA of the UHPLC-HRMS raw files. Chromatographic peaks of interest were aligned with a maximum shift of 0.1 minutes in retention time and a mass tolerance of $\pm 1$ ppm.

Ions were detected if the peak intensity was at least $5 \times 10^5$ counts for one of the following ions: $[\mathrm{M} - \mathrm{H}]^-$, $[\mathrm{M} - \mathrm{CO}_2 - \mathrm{H}]^-$, $[\mathrm{M} - \mathrm{H}_2\mathrm{O} - \mathrm{H}]^-$, and $[2\mathrm{M} - \mathrm{H}]^-$. In addition to the mass-to-charge ratio of the detected ion, at least one corresponding isotopologue has to be measured. The tolerance between measured and calculated intensity of the isotopologue has to be less than 30 %. Unknown compounds were then grouped with a retention time tolerance of 0.1 minute and those of them with a sample-to-blank ratio smaller than five were marked as background. A predicted composition was calculated within $\pm 1$ ppm,

allowing the elements carbon (C), hydrogen (H), bromine (Br), chlorine (Cl), nitrogen (N), oxygen (O), and sulfur (S). Compounds were grouped together as CHO, CHNO, CHOS, CHNOS, and "other" if the elemental composition contained other heteroatoms. For unidentified compounds the software does not predict a composition under the given conditions. The detailed workflow is given in Table S3.

To be clear on the degree of certainty regarding compound identification, we used the confidence levels from Schymanski

et al. (2014). Probably and tentatively labeled compounds correspond to level 2 and level 3, respectively. We used the mzCloud database (HighChem LLC, 2013-2021) for comparing MS$^2$ spectra of commercial chemicals. Unambiguously identified compounds (reference standard, MS$^2$ spectrum) correspond to confidence level 1.

Based on the Compound Discoverer results from the OFR experiments, we created a library for every examined chemical system (e.g., limonene and ozone) using mzVault 2.3 (Thermo Fisher Scientific Inc.), resulting in total 14 libraries of the

*aerosolomics* database. Every entry in a library contains the exact mass-to-charge ratio, the retention time, the MS$^2$ spectrum, and the relative abundance to the major product of the respective system if the relative abundance is higher than 1 %. These libraries were implemented in Compound Discoverer and aligned with the identified compounds from the representative selection of the field campaign. An entry in the library was dedicated to a compound in the ambient air filter samples if the difference in the retention times was smaller than 0.2 minutes, the measured mass-to-charge ratios of the ddMS$^2$ scans were

within a window of 10 ppm, and the match factor indicating the similarity of the MS$^2$ spectra was bigger than 50 %. Detailed settings of this node are given in Table S3. If a compound appears in several libraries, the match factor was crucial for an assignment. If match factors were equal, the system in which the compound has the highest relative abundance, was chosen.

We calculated the effective saturation mass concentration ($\log_{10} C^*$) for each compound with a predicted composition including at least the elements C, H, and O as well as N and S, based on Li et al. (2016). However, we like to point out that this

parameterization comprises a large molecular corridor and thus leads to a wide range of $\log_{10} C^*$. A bias has been reported for nitrogen containing compounds (Isaacman-VanWertz and Aumont, 2021), but also for CHO compounds it appears to be biased. For example, $\log_{10} C^*$ of the atmospheric tracer 3-methyl-1,2,3-butanetricarboxylic ($\mathrm{C}_8\mathrm{H}_{12}\mathrm{O}_6$, MBTCA) results in 1.97 $\mu\mathrm{g}\,\mathrm{m}^{-3}$, while with SIMPOL.1 (Pankow and Asher, 2008) we find $\log_{10} C^*$ at 298 K being $-2.2\,\mu\mathrm{g}\,\mathrm{m}^{-3}$. However, this difference of four orders of magnitude is certainly an extreme case, as all oxygen atoms of MBTCA are a part of a carboxylic

acid functional group.




We performed a HCA with MATLAB R2020a (The MathWorks) based on the complete data set from the Vienna field campaign including the blank-corrected integrated sample peak areas. After $z$-transformation, we used an Euclidean distance metric and the Ward algorithm for computing the distance between the clusters. For the compound clusters of the HCA, an intensity-weighted mean of $\log_{10} C^*$ was calculated. The borders of the different volatility classes are given accordingly to

Schervish and Donahue (2020). Based on this, the organic compounds can be classified as volatile (VOC), intermediate volatile (IVOC), semi-volatile (SVOC), low volatile (LVOC), extremely-low volatile (ELVOC), and ultra-low volatile (ULVOC).

## 3 Results and discussion

### 3.1 Oxidation flow reactor

OFR experiments provided SOA from several individually studied VOCs under different oxidation conditions. NTA of UHPLC-

HRMS measurements of the SOA extracts enabled us to populate our *aerosolomics* database with individual oxidation products of the studied VOCs in a qualitative way. We investigated five BVOCs and four AVOCs, and identified 481 and 115 oxidation products, respectively. Each of these 596 oxidation products is listed in the database with the information on precursor, oxidation condition, exact mass-to-charge ratio, retention time, MS$^2$-spectrum, and relative abundance in the respective system. Although, we are not able to determine the individual chemical structure of the different SOA compounds, their individual

structure leads to compound-specific retention times. Using all these parameters in the presented database allows unambiguous attribution of SOA compounds in ambient samples to their major parent VOC.

### 3.1.1 SOA compounds from oxidation of biogenic VOCs

We investigated the composition of SOA from the atmospherically most abundant biogenic monoterpenes ($C_{10}H_{16}$) $\alpha$-pinene, $\beta$-pinene, limonene, and 3-carene. The results are shown as mirror spectra in Fig. 2. The upper half of each subplot shows

the ozonolysis products under dark conditions, while the lower spectra show the products from OH oxidation (254 nm UV). Compounds with the five highest intensities are labeled with the predicted formula and their retention time, however, the database contains these entries of all compounds down to 1 % relative peak intensity.

Panel (a) shows the results of the $\alpha$-pinene oxidation experiments. Monomers produced during ozonolysis are mainly in the mass range between 140 and 210 Da, dimers are in the range between 300 and 400 Da. The major products during ozonol-

ysis are $C_9H_{14}O_4$ (at 8.79 min), $C_8H_{12}O_4$ (at 6.67 min), $C_{17}H_{26}O_8$ (at 11.28 min), $C_8H_{14}O_5$ (at 5.84 min), and $C_8H_{14}O_6$ (at 6.56 min). Oxidation by OH reduces the absolute signal intensity of most oxidation products (see Fig. S2). Furthermore, this oxidation environment prevents the production of dimers and certain monomers, and changes the relative abundance of several monomers. For example, the relative abundance of pinic acid ($C_9H_{14}O_4$ at 8.79 min, level 1), which is the major compound of the ozonolysis, decreases by 30 % under OH conditions. In contrast, the relative abundance of several other

compounds increase, which indicates a higher relevance in the OH system, like $C_8H_{12}O_4$ (at 6.67 min) increasing to 100 % relative abundance, $C_{10}H_{16}O_5$ (at 9.28 min) increasing to 88 % relative abundance, $C_8H_{14}O_5$ (at 5.84 min) increasing from



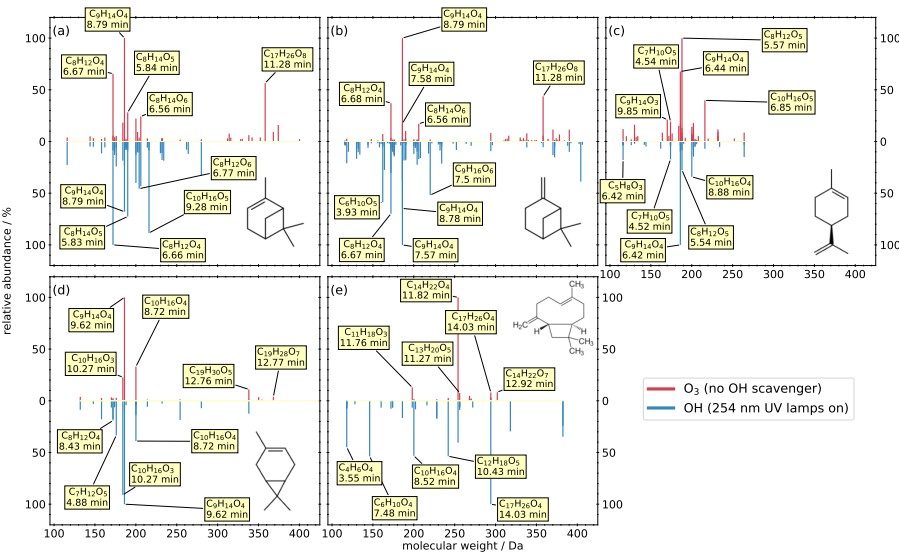

**Figure 2.** Mass spectra of the detected products from the OFR experiments of five biogenic precursors (a) $\alpha$-pinene, (b) $\beta$-pinene, (c) limonene, (d) 3-carene, (e) and *trans*-caryophyllene. The intensity is normalized to the highest signal of each chemical system. The five most intensive compounds of each experiment are labeled with their predicted composition and the according retention time.

28 % to 73 %, and $C_8H_{12}O_6$ (at 6.77 min) increasing from 4 % to 46 %. Panel (b) shows the results of the $\beta$-pinene oxidation experiments. Here, many compounds are similar to the $\alpha$-pinene oxidation products, with the exception of dimer formation during OH conditions. For both ozonolysis and OH oxidation, $C_9H_{14}O_4$ is the compound with the highest relative abundance,

200   although the chromatography resolves different isomers: In the upper spectrum ($O_3$) the isomer at 8.79 minutes has a relative abundance of 100 %, whereas the isomer at 7.58 minutes has a relative abundance of 17 %. In the lower spectrum (OH), the relative abundances are reversed, with 100 % at 7.57 minutes and 65 % at 8.78 minutes. This indicates that different oxidation conditions of the same precursor result in different isomers of $C_9H_{14}O_4$, which can only be resolved with chromatographic separation. Furthermore, the isomer at 7.57 minutes does not appear in any other experiment in higher amounts, for which rea-

205   son it can be used as a specific $\beta$-pinene tracer. While the most prominent dimer $C_{17}H_{26}O_8$ (at 11.28 min) appears analogously to the $\alpha$-pinene-system during ozonolysis, here, under OH conditions, $\beta$-pinene oxidation results in dimer oxidation products like $C_{19}H_{32}O_9$ (at 13.23 min), in contrast to no dimers in $\alpha$-pinene-system.

Panel (c) shows the results of the limonene oxidation experiments. In contrast to the other three monoterpenes no dimers were formed, which is in general agreement with Hammes et al. (2019). The ozonolysis shows three major products, $C_8H_{12}O_5$

210   (at 5.57 min), $C_9H_{14}O_4$ (at 6.44 min), and $C_{10}H_{16}O_5$ (at 6.85 min). In the OH system $C_9H_{14}O_4$ (at 6.44 min) becomes the major compound whereas the intensity of $C_8H_{12}O_5$ (at 5.57 min) increases clearly. Analogous to the $\beta$-pinene oxidation, the $C_9H_{14}O_4$ isomer at 6.44 minutes can be used as specific limonene tracer due to the missing appearance of this isomer in other experiments.





Panel (d) shows the results of the 3-carene oxidation experiments. Three monomers are the most prominent products in both systems: $C_9H_{14}O_4$ (at 9.62 min), $C_{10}H_{16}O_4$ (at 8.72 min), and $C_{10}H_{16}O_3$ (at 10.27 min). The four dimers $C_{19}H_{30}O_5$ (at 12.76 min), $C_{19}H_{28}O_7$ (at 12.77 min), $C_{20}H_{30}O_5$ (at 13.74 min), and $C_{19}H_{28}O_9$ (at 11.40 min) appear during ozonolysis from which three are also reported by Thomsen et al. (2021), tentatively identified as dimers from 3-carene. Under OH conditions the dimers disappear with the exception of $C_{19}H_{30}O_5$.

In addition to the four monoterpenes, we investigated the composition of sesquiterpene-SOA from *trans*-caryophyllene ($C_{15}H_{24}$). During ozonolysis we find one major and four minor products with molecular weights in the mass range between 198 and 302 Da (Fig. 2e). The major compound is tentatively identified as $\beta$–nocaryophyllonic acid ($C_{14}H_{22}O_4$ at 11.82 min, level 3) (van Eijck et al., 2013; Jaoui et al., 2003). In contrast, the reaction with OH leads to one major and seven minor products in a range from 118 Da up to 382 Da. The major compound $C_{17}H_{26}O_4$ (at 14.03 min) also appears during ozonolysis but only with a relative abundance of 8 %.

Considering BVOC oxidation in general, it is worth mentioning that different isomers of $C_9H_{14}O_4$ are clearly separated by the chromatographic system and we can use them as specific tracers for different BVOCs in the *aerosolomics* database. These and even more isomers are present in ambient filter samples (Fig. S3), which demonstrates the necessity of chromatographic separation if an unambiguous assignment is desired. Furthermore, ion source dimerization is a known phenomenon that hinders the unambiguous identification of atmospheric dimers, or leads to misinterpretation of results from direct-injection HESI. Based on the knowledge of the exact *m/z* and the mass dependence of the retention time, we can assign ion source related dimers to the associated atmospheric monomer. This allows us an unambiguous distinction between monomers and covalently bonded "real" dimers (Fig. S4).

### 3.1.2 SOA compounds from oxidation of anthropogenic VOCs

We investigated the composition of SOA from the anthropogenic VOCs 1,2,4-trimethylbenzene, toluene, *o*-xylene, and naphthalene. We carried out only OH-oxidation of AVOCs, because oxidation of aromatic compounds by $O_3$ is negligible. The filter criteria were similar to the experiments with BVOCs and the resulting spectra are shown in Fig. 3. All experiments show a noticeably lower number of oxidation products compared to biogenic precursors. We observe dimers only in the 1,2,4-trimethylbenzene and the *o*-xylene system.

Panel (a) of shows the results of the 1,2,4-trimethylbenzene oxidation. The two most prominent compounds are $C_9H_8O_4$ (at 9.04 min) and $C_5H_6O_4$ (at 4.29 min). The remaining compounds play a minor role due to relative abundances less than 10 %.

Panel (b) shows the results from the oxidation of toluene. The five most prominent compounds show a higher relative abundance than 40 % and no compound has a lower abundance than 15 %. Most of these products are highly oxygenated with more than four oxygen atoms with the exception of the highest signal corresponding to $C_5H_6O_3$. All the small (C $\leq$ 5) highly oxygenated molecules exhibit also a high polarity ($t_R < 2$ min) compared to the oxidation products of other anthropogenic precursors.



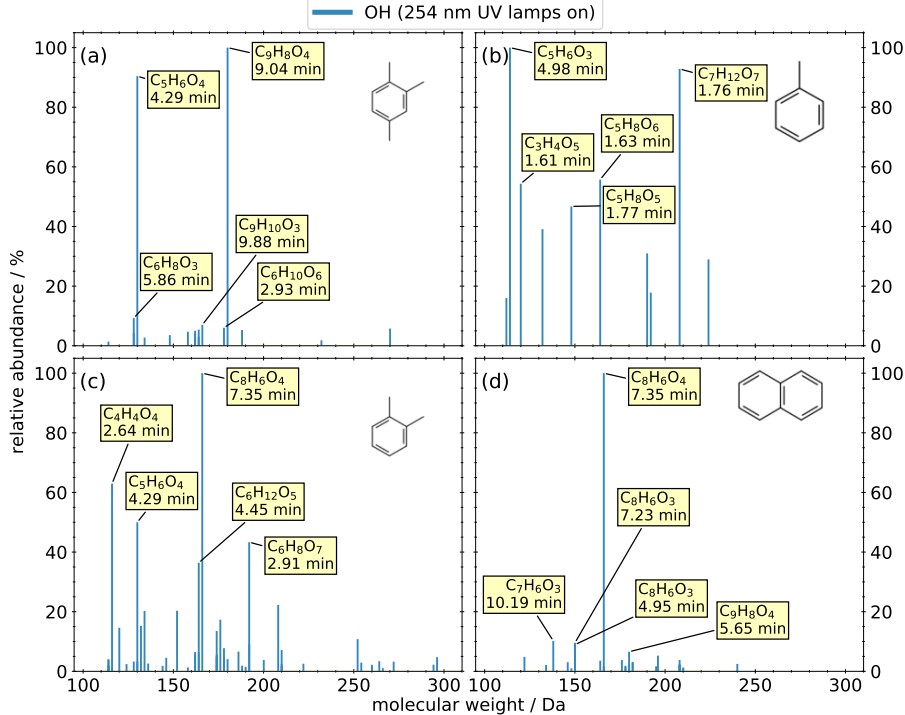

**Figure 3.** Mass spectra of detected products from OFR experiments of four anthropogenic precursors (a) 1,2,4-trimethylbenzene, (b) toluene, (c) $o$-xylene, (d) and naphthalene. The intensity is normalized to the highest signal of each chemical system. The five most intensive compounds of each experiment are labeled with their predicted composition and the according retention time.

Panel (c) shows the results of the $o$-xylene oxidation, the anthropogenic precursor with the largest number of detected oxidation products (n = 52) of the four investigated AVOCs. While no composition could be assigned by the NTA software for the highest signal, due to an invalid isotopic pattern, the most abundant product in panel (d) appears on the same mass trace and the same retention time. This peak was identified as phthalic acid ($C_8H_6O_4$, level 1), which is described as naphthalene SOA tracer by Al-Naiema et al. (2020).

Panel (d) shows the oxidation of naphthalene resulting in the main oxidation product phthalic acid. All other compounds have a relative abundance smaller than 20 %. It is worth mentioning that two isomers of $C_8H_6O_3$ appear with a similar relative abundance, but with two distinguishable retention times.

## 3.2  *Aerosolomics*-database application on ambient samples

### 3.2.1  Fingerprint

The NTA of the representative selection of the Vienna field campaign extracts results in 1312 compounds shown in panel (a) of Fig. 4 as retention time as a function of molecular weight. The scatter size represents the mean signal intensity of the 10 mea-





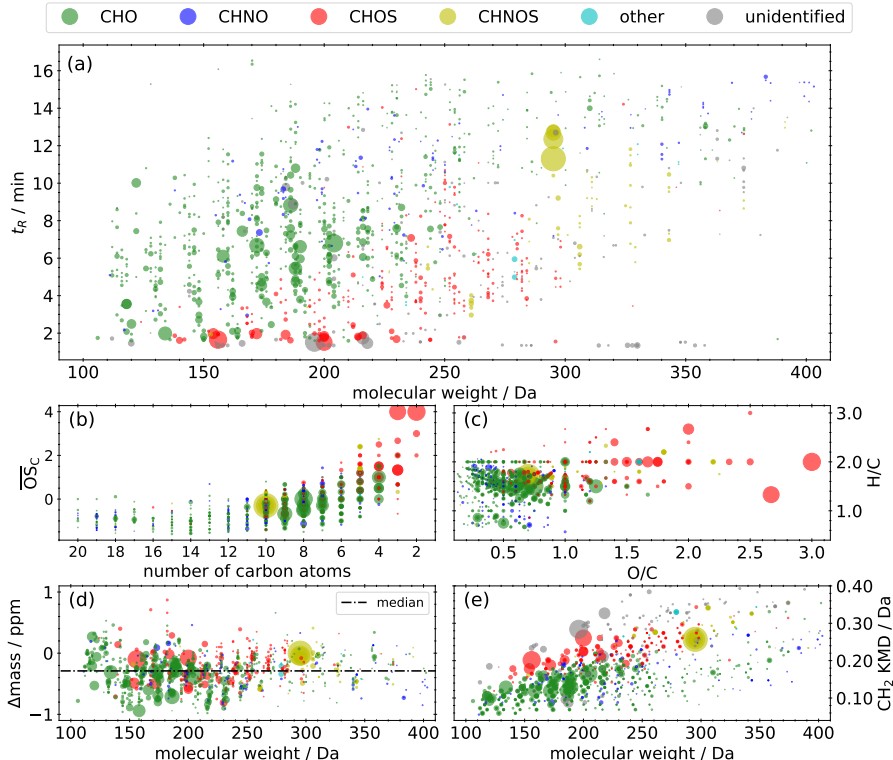

**Figure 4.** Molecular fingerprint of the representative selection of the field campaign samples. The area of the scatters represents the qualitative mean signal intensity on a linear scale. (a) Retention time vs. molecular weight. (b) Kroll diagram. (c) Van Krevelen diagram. (d) Mass difference between calculated and measured molecular weight ($\Delta$mass) vs. molecular weight. (e) Kendrick mass defect plot. Panels (b), (c), and (d) only include compounds with a predicted composition.

surements on a linear scale. The majority are CHO compounds (54.7 %), followed by CHOS (17.3 %), CHNOS (13.1 %), and unidentified (12.3 %) compounds (Fig. 5) spanning the range from $100\,\text{Da}$ to $350\,\text{Da}$. CHNO and other compounds, especially halogenated hydrocarbons, play a minor role with proportions of 2.2 % and 0.4 %, respectively. The entire range of retention times indicates a broad mixture of polarities. Among the most common signals are nitrooxy-organosulfates like $C_{10}H_{17}NO_7S$, isoprene-derived CHOS like $C_2H_4O_6S$ (Claeys and Maenhaut, 2021) as well as monoterpene oxidation products like $C_8H_{12}O_6$ (terpenylic acid, level 3), $C_8H_{12}O_4$ (MBTCA, level 1), and $C_9H_{14}O_4$ (pinic acid, level 1).

Beside the two series of isoprene-derived CHNOS isomers highlighted in Sect. 3.3.1, three additional series of isomers can be found in the molecular fingerprint: $C_{10}H_{17}NO_7S$ ($295\,\text{Da}$), $C_{10}H_{17}NO_9S$ ($327\,\text{Da}$), and $C_{10}H_{17}NO_{10}S$ ($343\,\text{Da}$) are all described as monoterpene derived SOA (Surratt et al., 2008). Especially $C_{10}H_{17}NO_7S$ plays an prominent role, due to the overall maximum intensity, and it illustrates the anthropogenic influence on the oxidation products from BVOCs in the presence of $NO_x$ and $SO_2$. The appearance of the ions $NO_3^-$ and $HSO_4^-$ in the $MS^2$ spectra prove that these compounds are nitrooxy





270 organosulfates. Overall, 86 % of the CHNOS compounds show both ions in their MS$^2$ spectra and can also be attributed to this group.

The average carbon oxidation state ($\overline{\mathrm{OS}}_\mathrm{C}$) according to Kroll et al. (2011) is in the range between $-1.5$ and 3, with the exception of two small CHOS compounds with an $\overline{\mathrm{OS}}_\mathrm{C}$ of up to 4 (Fig. 4b). The majority of the CHO compounds consist of equal to or less than ten carbon atoms, while the majority of the CHOS compounds consist of equal to or less than five carbon 275 atoms. This pattern indicates the importance of monoterpenes and isoprene as SOA precursors.

The Van Krevelen diagram (Fig. 4c) clarifies the influence of biogenic precursors. On the one hand the majority of the CHO compounds appear in the H/C range between 1.2 and 1.8, described as a surrogate for biogenic SOA (Daellenbach et al., 2019; Kourtchev et al., 2015). On the other hand, compounds with a H/C < 1.2, indicating aromatic character, are rare despite proximity to an airport and a large refinery. Even though, several CHNO tracers for biomass burning could be identified, they only play 280 a minor role with regard to noticeably lower signal intensities: $C_7H_5NO_5$ (nitrosalicylic acid, level 3), $C_7H_7NO_4$ (methylnitrocatechol, level 3), $C_7H_6N_2O_5$ (methyldinitrophenol/dinitrocresol, level 2), $C_7H_6N_2O_6$, $C_6H_5NO_4$ (nitrocatechol, level 2), $C_6H_4N_2O_5$ (dinitrophenol, level 2), $C_8H_7NO_4$ (methylnitrobenzoic acid, level 3), and $C_8H_9NO_5$ (Ikemori et al., 2019; Mohr et al., 2013; Salvador et al., 2021).

In order to evaluate optimal settings for chemical composition prediction during NTA and to avoid false predictions, the 285 mass difference between calculated and measured molecular weight ($\Delta$mass) has to be considered. Figure 4d shows that over the entire mass range only a slight shift to negative values (median $= -0.29$ ppm) can be observed within the space between $-1$ and 1 ppm deviation. Within the set of allowed elemental compositions an enlargement of the range ($\pm 1$ ppm) would increase false predictions due to the increased mathematical possibility of other elemental combinations. The Kendrick mass defect (KMD) plot (Fig. 4e) allows the identification of homologous series. Members of such series have the same Kendrick 290 mass defect (Kendrick, 1963) like $C_nH_{2n-2}O_3$ with n $=5$–15 (KMD $=0.082$), $C_nH_{2n-4}O_4$ with n $=4$–16 (KMD $=0.119$), or $C_nH_{2n-2}O_5$ with n $=3$–14 (KMD $=0.128$). Also CHOS compounds form homologous series like $C_nH_{2n}O_5S$ with n $=2$–8 (KMD $=0.178$), or $C_nH_{2n}O_6S$ with n $=2$–10 (KMD $=0.201$). The appearance of these homologous series can be interpreted as oxygenated aliphatic hydrocarbons of fossil origin, as homologous series of (sulfur-containing) aliphatics can be detected in crude oil.

### 295    3.2.2   *Aerosolomics* database assignment

The results from the NTA of the ambient samples were compared with the outcome from the PAM-OFR experiments in order to estimate the contribution of oxidation products formed by several precursors to ambient PM$_{2.5}$. Out of a total of 580 detected CHO compounds, we can assign 108 CHO compounds to biogenic precursors and 24 CHO compounds to anthropogenic VOCs using our *aerosolomics* database. The bar plot in Fig. 5 divides the CHO fraction into the different contributions 300 examined. 40.8 % of the mean signal intensity can be attributed to SOA originated from biogenic precursors (gradations of green and yellow). Out of these, 26.5 % account for experiments with OH oxidation and 14.3 % for ozonolysis experiments. The two major biogenic precursors are $\alpha$- and $\beta$-pinene with a respective share of 17.6 and 13.2 %. The remaining shares are distributed among *trans*-caryophyllene products (4.2 %), among limonene products (3.6 %), and among 3-carene products





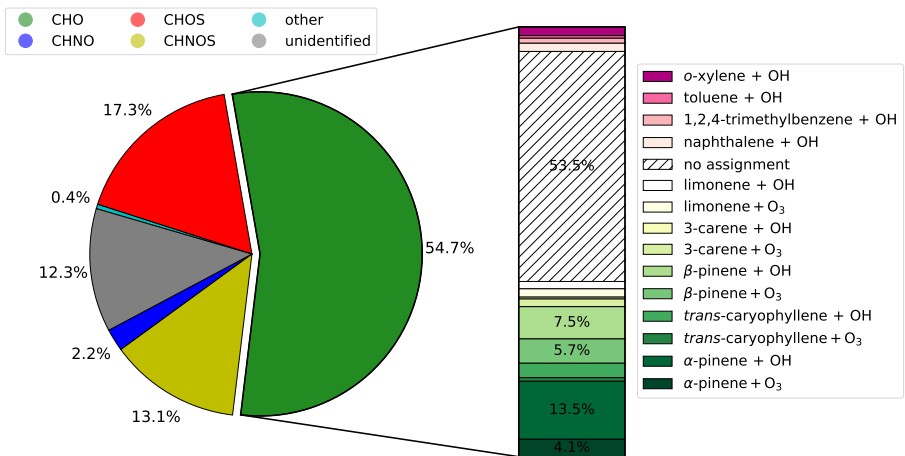

**Figure 5.** Contribution of different molecular formula groupings to the mean signal intensity of the molecular fingerprint from the representative selection shown in Fig. 4. Compounds of the CHO group were compared to the libraries of the individual OFR experiments presented in Sect. 3.1 and dedicated to a plausible SOA precursor and oxidation condition shown in the bar.

(2.2 %). Anthropogenic precursors (gradations of magenta) play a minor role with a total contribution of 5.7 %. From that 2 %

can be assigned to *o*-xylene, 1.9 % to naphthalene, 1.2 % to 1,2,4-trimethylbenzene, and 0.6 % to toluene. 53.5 % of the mean signal intensity could not be assigned (diagonally hatched).

The majority of the assigned compounds from the representative selection of the Vienna field campaign have a molecular weight smaller than 250 Da and a retention time lower than 10 minutes (Fig. S5a). The mean number of carbon atoms is 9 and the mean $\overline{OS}_C$ is $-0.4$ (Fig. S5b). SOA originated from biogenic precursors is located in the H/C area between 1.2 and 1.8

(Fig. S5c), while compounds with H/C < 1.2 can be of aromatic character. The observation that 19 % of the CHO compounds (number-wise) in the database are responsible for nearly 50 % of the mean signal intensity demonstrates the high relevance of the investigated VOCs in SOA formation. Nevertheless, a few compounds with high signal intensities remain unassigned. Considering the molecular weight of these unassigned compounds we expect isoprene as well as other monoterpenes to be promising candidates closing this gap. A comprehensive study of isoprene oxidation is planned and the outcome will be

uploaded to the *aerosolomics* database in the near future.

Beside CHO compounds, CHOS and CHNOS compounds play an important role in the overall composition of suburban SOA. Based on these outcome, further experiments with various VOCs and complex mixtures including inorganic trace gases need to be performed and the results have to be added to the *aerosolomics* database.

### 3.3  Hierarchical cluster analysis

In Fig. 6 we show the results of the HCA (as a heatmap with dendrograms), with the ambient PM$_{2.5}$ filter samples from the Vienna field campaign on the horizontal axis, and the detected compounds of the NTA on the vertical axis. The color code of the heatmap represents the standardized values of the integrated peak intensities after background correction. We find that





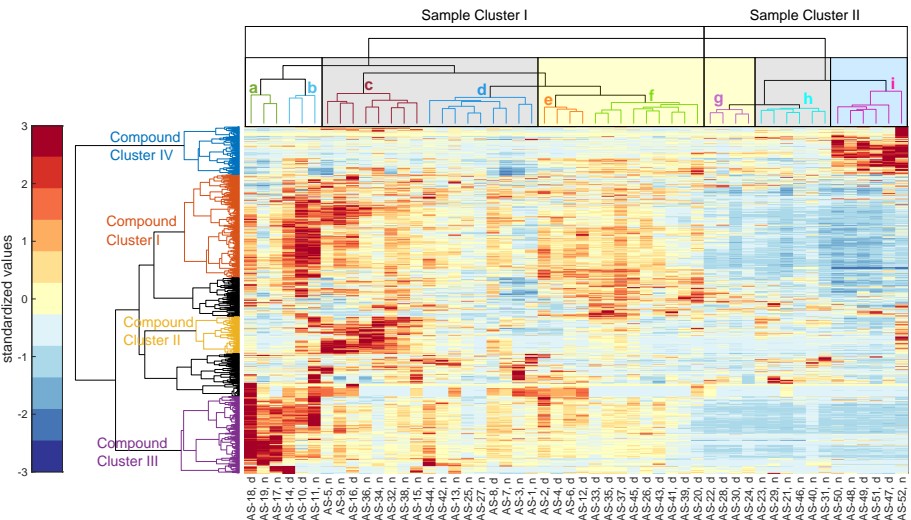

**Figure 6.** Standardized matrix of the detected compounds from the NTA of the filter samples from the Vienna field campaign. Parts of the horizontal dendrogram of the filter samples are shaded in grey for night cluster, yellow for day cluster and blue for clustered samples at decreasing temperatures. The vertical compound clusters are separated into four subclusters according to day- and night-time chemistry, organosulfates, and temperature dependency. For a better differentiation the dendrograms are coloured randomly.

the driving factors for the clustering of the filter samples are the wind direction overlaid by the diurnal cycle, as well as the influence of decreasing temperatures, as explained in the following section. Clusters that contain mostly night-time samples are shaded by a grey background. The ones that contain mainly daytime samples are shaded by a yellow background, while samples during a low-temperature period are shaded by a blue background. Over the entire period the mean $PM_{2.5}$ mass concentration was $8.7 \pm 4.4\,\mu g\,m^{-3}$, the mean $NO_x$ concentration was $15.4 \pm 16.7\,\mu g\,m^{-3}$, the mean $SO_2$ concentration was $1.5 \pm 1.2\,\mu g\,m^{-3}$, and the mean CO concentration was $0.17 \pm 0.03\,mg\,m^{-3}$. A detailed time series of meteorological data, $PM_{2.5}$, and trace gas concentrations are shown in Fig. S6. Additionally, Fig. S7 shows the distribution of the wind direction of sample clusters a to i. With exception of sample clusters b and f, all clusters show a predominant wind direction. The overall mean $PM_{2.5}$ mass concentration was higher during collection in sample cluster I ($10.3 \pm 4.1\,\mu g\,m^{-3}$) compared to sample cluster II ($5.0 \pm 2.4\,\mu g\,m^{-3}$).

### 3.3.1 Compound cluster I – daytime chemistry

Figure 7a–c illustrates the molecular fingerprint of the 373 compounds occurring in compound cluster I. 79 % of the mean signal intensity is caused by CHO compounds, 16 % is unidentified, and 4 % is caused by CHOS compounds. The molecular weight of the compounds are in the mass range between 100 and 350 Da. CHO compounds have mainly weights smaller than 250 Da and a mean bulk composition of $C_{8.5}H_{12.5}O_5$ which is in good agreement with the appearance of monomers from monoterpene oxidation during daytime. The number of carbon atoms ranges mainly between 4 and 10 (Fig. 7b), indicating biogenic VOCs, like monoterpenes, as potential precursors. The compounds cover a large range in volatility, with $\log_{10} C^*$





values between $-4.0$ and $6.5\,\mu g\,m^{-3}$ (Fig. 8a), corresponding mainly to LVOC–IVOC with an intensity weighted $\log_{10} C^*$ mean value of $2.3 \pm 1.6\,\mu g\,m^{-3}$. Only a small fraction of the detected compounds have a H/C smaller 1.2, indicating an aromatic character. The five most intense compounds with a predicted composition are $C_8H_{12}O_6$ (MBTCA, level 1, *aerosolomics* marker: $\alpha$-pinene + OH), $C_8H_{12}O_5$, $C_5H_6O_7$, $C_4H_6O_5$, and $C_7H_{10}O_5$. All five compounds are characterized as biogenic SOA-compounds derived from isoprene or monoterpenes (Chen et al., 2018, 2020; Ehn et al., 2012; Müller et al., 2012; Qi

et al., 2020).

CHOS compounds appear less important in this compound cluster, based on a 5 % contribution to the mean signal intensity. Nevertheless, about 45 % of the mean CHOS signal intensity can be attributed to monoterpene and isoprene derived SOA (Brüggemann et al., 2020).

It is also remarkable that compounds appearing in this cluster show low standardized values in sample cluster II. Low

standardized values indicate lower signal intensities of the MS measurements. However, it need to be considered that signal intensities are not directly quantitative to its concentration. It is well known, that the ionization efficiency of HESI varies greatly for several compounds as well as compound classes (Kenseth et al., 2020; Ma et al., 2022). Nevertheless, variation in the intensity of a single compound or class of compounds can be qualitatively interpreted as variation of its concentration. Compared to a mean temperature of $25.5 \pm 4.8\,°C$ during sample cluster I, the mean temperature of sample cluster II is notice-

ably lower with $19.3 \pm 4.3\,°C$. Especially the subcluster including the last six samples of the entire field campaign shows the lowest standardized values and the lowest mean temperature of $16.1 \pm 3.3\,°C$. The correlation of lower temperatures and low standardized values can be explained due to the temperature dependency of terpene emissions from plants (Holzke et al., 2006) resulting in a lower biogenic SOA burden in the atmosphere.

### 3.3.2   Compound cluster II – night-time chemistry

The mean signal intensity of the 134 compounds appearing in compound cluster II (Fig. 7d–f) is mainly caused by CHO (78 %) and CHNO (14 %) compounds. The molecular weight reaches up to $440\,Da$ and the mean bulk composition is $C_{11.9}H_{18.2}O_{5.1}$. The $\overline{OS}_C$ of these CHO compounds are in the range between $-1$ and 0.5 (Fig. 7e). Analogously to compound cluster I, the H/C is in the range between 1.2 and 1.8 (Fig. 7f), indicating biogenic SOA. $\log_{10} C^*$ values range from $-7.1$ to $6.7\,\mu g\,m^{-3}$ (Fig. 8b) corresponding mainly to LVOC–IVOC with an intensity weighted mean of $2.3 \pm 2.7\,\mu g\,m^{-3}$. The most prominent

compound is $C_9H_{14}O_4$ (pinic acid, level 1, *aerosolomics*-database library: $\alpha$-pinene/$\beta$-pinene + O$_3$) with a contribution of about 24 % to the mean signal intensity. It is well described as a $\alpha$- and $\beta$-pinene ozonolysis product (Christoffersen et al., 1998; Glasius et al., 2000).

In contrast to sample cluster I, night chemistry related compounds form a higher ratio of dimers with oxidation products up to 20 carbon atoms (Fig. 7e). 14 of 23 CHO dimers occuring in compound cluster II are also reported from OFR experiments

performed by Kristensen et al. (2016) and agree with our findings from the OFR experiments described in Sect. 3.1. These 14 dimers are responsible for 72 % of the mean dimer signal intensity. Among them are $C_{17}H_{26}O_8$ (*aerosolomics*-database library: $\alpha$-pinene/$\beta$-pinene + O$_3$) and $C_{19}H_{28}O_7$ (*aerosolomics*-database library: $\alpha$-pinene/3-carene + O$_3$), both of which are





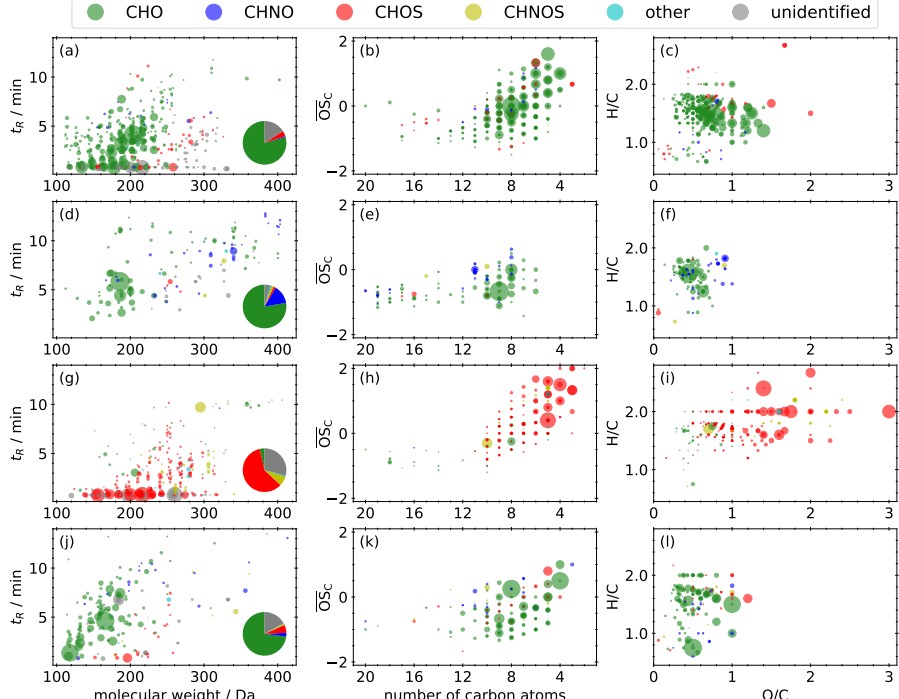

**Figure 7.** Molecular fingerprint of compounds that appear in compound cluster I (a–c), II (d–f), III (g–i), IV (j–l), illustrated as retention time vs. molecular weight, Kroll diagram, and Van Krevelen diagram. The area of the scatters represent the mean signal intensity on a linear scale. The pie chart shows the proportions of the different molecular formula groupings.

described as esters between pinic acid and terpenylic acid (Gao et al., 2010) or pinic acid and hydroxypinonic acid (Müller et al., 2008), respectively.

While during daytime the main oxidant of VOCs is OH, at night-time $O_3$ and $NO_3$ dominate the oxidation process, resulting in the formation of organonitrates (Kiendler-Scharr et al., 2016). In accordance, we observed several nitrogen containing monomers and dimers including $NO_3^-$ in the corresponding $MS^2$ spectra. Hence, about $87\%$ of the CHNO compounds are tentatively identified as organonitrates ($R-ONO_2$).

### 3.3.3   Compound cluster III – organosulfates (CHOS)

Compound cluster III includes 279 compounds (Fig. 7g–i). The mean signal intensity of compound cluster III is mainly caused by CHOS compounds ($60\%$), followed by unidentified ($29\%$) and CHNOS compounds ($7\%$). The molecular weight of the compounds reaches from 120 to $350\,\mathrm{Da}$, those with higher mean signal intensities show only a very limited retention ($<0.9\,\mathrm{min}$) due to high polarity and represent about $72\%$ of the mean signal intensity. These compounds have $\leq 5$ carbon atoms and an $\overline{OS}_C$ between 0 and 4 (Fig. 7h). The generally higher H/C compared to other compound clusters, shown in panel (i), indi-





cate a predominantly aliphatic character of the detected compounds. The saturation vapor pressure ($\log_{10} C^*$) ranges between $-8.8$ and $5.0\,\mu\mathrm{g\,m^{-3}}$ (Fig. 8c), which is noticeably lower compared to compound clusters I and II. The intensity-weigthed mean of $\log_{10} C^*$ of this cluster yields $-0.9 \pm 2.0\,\mu\mathrm{g\,m^{-3}}$.

Several of the chemical formulas can be described as isoprene-derived SOA (Brüggemann et al., 2020; Chen et al., 2018; Nestorowicz et al., 2018; Riva et al., 2016; Surratt et al., 2007). From 174 CHOS compounds in compound cluster III, 163

MS$^2$ spectra were recorded. From those the vast majority (98 %) shows the fragment at *m/z* 96.9601 ($\mathrm{HSO_4^-}$), indicating an organosulfate functional group ($\mathrm{R-OSO_3}$). The remaining 2 % only show *m/z* 79.9573 ($\mathrm{SO_3^{\bullet-}}$). While this sulfur trioxide radical anion can indeed occur in an organosulfate fragment spectra (Wang et al., 2019), it could also be originated from organosulfonates ($\mathrm{R-SO_3}$) (Liang et al., 2020; Liu et al., 2015).

Beside one high signal intensity of a CHNOS compound at 295 Da and 9.7 minutes ($\mathrm{C_{10}H_{17}NO_7S}$), three further series of

isomers are appearing in the CHOS-cluster: We identified six isomers of $\mathrm{C_5H_{10}N_2O_{11}S}$, four isomers of $\mathrm{C_5H_{11}NO_9S}$, and three isomers of $\mathrm{C_5H_9NO_8S}$. All three chemical formulas can be described as isoprene-derived SOA (Nestorowicz et al., 2018; Surratt et al., 2007, 2008). Furthermore, three isomers of $\mathrm{C_5H_9NO_7S}$ also appear in this compound cluster, but has not yet been described as isoprene-derived SOA.

The distinct increased standardized values of several Cluster-III-compounds in the heatmap between 8. August 2018, 17:00

(UTC, AS-17) and 10. August 2018, 05:00 (UTC, AS-19) can be explained by high $\mathrm{SO_2}$ concentrations up to $18\,\mu\mathrm{g\,m^{-3}}$ around noon on 8 August 2018 (Fig. S6). The nearby airport as a main source can be excluded due to similar high $\mathrm{SO_2}$ concentrations at the Stixneusiedl monitoring station (Umweltbundesamt GmbH, 2021) located 12 km southeast of the airport and thus in the upwind direction of the airport. It is also noticeable, that the standardized values of these Cluster-III-compounds are very low during northwest trajectories, analogously to compound cluster I (Sec. 3.3.1). Therefore, it is likely that this cluster is mainly

linked to long-range transport of pollution from the south-east.

### 3.3.4 Compound cluster IV – decreasing temperature

Compounds occurring in compound cluster IV (Fig. 7j–l) show clearly increased standardized values during the last six filter samples of the field campaign and are mainly CHO (73 %) and unidentified (17 %) compounds. The majority of the Cluster-IV-compounds have a molecular weight $< 200\,\mathrm{Da}$. The $\overline{\mathrm{OS}}_\mathrm{C}$ of those compounds ranges between $-1.5$ and $1$ while the num-

bers of carbon is less than 10 (Fig. 7k). The majority of the 174 compounds have a non-aromatic character, illustrated in panel (l). Dicarboxylic acids, like phthalic acid ($\mathrm{C_8H_6O_4}$, level 1), succinic acid ($\mathrm{C_4H_6O_4}$, level 3), or maleic/fumaric acid ($\mathrm{C_4H_4O_4}$, level 3) are reported as tracers for emissions from biomass burning, vehicular exhaust and fossil fuel combustion (Zhao et al., 2018). Furthermore, this compound cluster contains the homologous series $\mathrm{C_{4-9}H_{6-16}O_4}$, $\mathrm{C_{5-10}H_{8-18}O_3}$, and $\mathrm{C_{5-9}H_{10-18}O_3}$, which can be interpreted as oxygenated aliphatics of fossil origin.

Compound cluster IV includes $\log_{10} C^*$-values between $-8.6$ and $6.5\,\mu\mathrm{g\,m^{-3}}$ (Fig. 8d) corresponding predominantly to SVOC–IVOC with an intensity weighted mean of $3.9 \pm 1.9\,\mu\mathrm{g\,m^{-3}}$. The high standardized values in the dendrogram of these compounds in sample subcluster i can be attributed by their intermediate volatility, which will only occur in the condensed



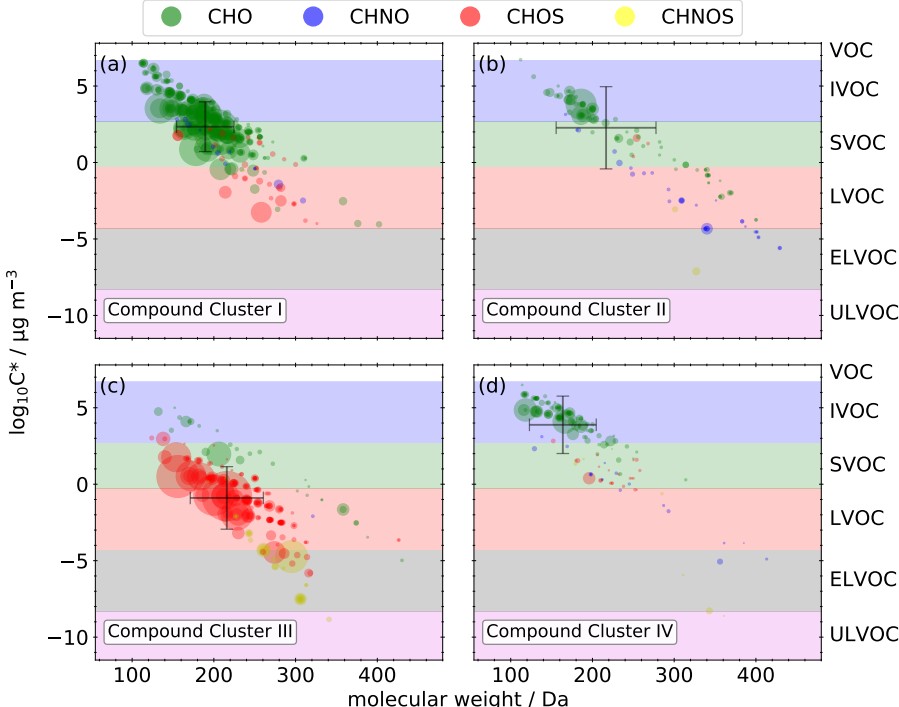

**Figure 8.** Calculated effective saturation mass concentrations of compounds appearing in compound cluster I-IV (a–d) with the intensity weighted mean and standard deviation.

particle phase at low ambient temperature. During sample cluster I, which is characterized by a higher mean temperature, those compounds would likely partition predominantly into the gas phase.

### 3.4 Challenges with measurements in HESI positive ionization mode

All results in this paper are based on negative ionization mode measurements. For a comprehensive assessment of the chemical composition of the investigated $PM_{2.5}$ measurements in positive ionization mode ((+)HESI) are essential with respect to relevant compound classes that are considerably better ionized in (+)HESI (e.g. organophosphates, phthalates, pesticides, and others). Issues due to strong fragmentation or ion-source adduct formation, which are explained in detail in the supplement, highlight the need of further optimization of the NTA workflow to prevent false identifications and thus misinterpretations of the results.

### 4 Conclusions

A large fraction of ambient $PM_{2.5}$ consists of anthropogenic and biogenic SOA. The chemical composition of this mixture is highly complex, which hinders the identification and attribution of single molecules to its precursors, potential sources,



and formation pathways. In this study, we present two complementary approaches that enable complexity reduction, and identification of precursors, formation pathways, as well as partitioning effects of various IVOCs.

The introduced *aerosolomics* database for compound matching and precursor identification is based on PAM-OFR experiments with five biogenic and four anthropogenic precursors, as well as different oxidizing regimes (OH, $O_3$). In order to validate the method on ambient samples, we applied the database on a set of $PM_{2.5}$ filter samples. Based on the average

composition of these samples, we find that CHO compounds account for the largest proportion with about 55 % of the mean signal intensity. Approximately 45 % of the CHO compounds can be attributed to one of the investigated VOC precursors, which we confirmed by the compound matching procedure. Hence, the compounds are identified based on retention time, exact mass-to-charge ratio, isotopic pattern, and the $MS^2$ fragmentation spectrum.

On a one-month set of filter samples, we performed a HCA to reduce the complexity due to the large number of compounds

detected. The compounds were clustered when their intensities show similar behavior over time, which in turn indicates similar sources or (trans-)formation pathways. The clustering of the various $PM_{2.5}$ filter samples was primarily driven by wind direction, as well as by the diurnal cycle (day/night) and temperature-driven partitioning changes. Known proxies for monoterpene ozonolysis, like pinic acid ($C_9H_{14}O_4$) or $\alpha$-pinene-derived dimers, were identified in the night-time compound cluster. The SOA aging tracer MBTCA appeared in the daytime compound cluster. A large number of sulfur-containing compounds were

clustered together, and this cluster is clearly elevated during south-easterly wind direction. Small IVOCs were clustered and show a high intensity during a cold period. These observations are a proof-of-principle that (1) the presented *aerosolomics* database enables to identify tracers from the oxidation of different VOC precursors, and in combination with HCA we can (2) attribute different oxidation products to either night-time or daytime chemistry, (3) identify periods of multiphase-chemistry processes resulting in organosulfate formation, and (4) observe temperature-driven partitioning of IVOCs.

We would like to encourage the community to apply the database on their own samples. As a community effort, further input to the database is desirable to improve our understanding of sources and formation of secondary organic aerosol.

*Data availability.* Data from this work are freely available at zenodo.org. DOI: 10.5281/zenodo.6623244

*Author contributions.* MT was the main author, ALV and MS advised on manuscript writing. MT, FB, FG and MS were responsible for the laboratory experiments, sample preparation, and measurements. Data evaluation was done by MT. ALV directed the project administration.

All authors commented on the manuscript and contributed to the scientific discussion.

*Competing interests.* The authors declare that they have no conflict of interest.





*Financial support.* This work was supported by Emmy Noether-Programm (Deutsche Forschungsgemeinschaft), project number 410009325. This open-access publication was funded by the Goethe-University Frankfurt.

*Acknowledgements.* We thank Paul Winkler and Sophia Brilke (Faculty of Physics, University of Vienna), for ambient filter sampling and
trace gas monitoring. We also thank the working group of Joachim Curtius (Institute for Atmospheric and Environmental Sciences, Goethe-University Frankfurt) for experimental support.



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
