# Peer review of "Mass spectrometry-based *aerosolomics*: a new approach to resolve sources, composition, and partitioning of secondary organic aerosol"

_Atmospheric Measurement Techniques, 2022_

## Referee Comment (RC2)

Thoma et al. conducted laboratory experiments in which they generated SOA from O3/OH oxidation of biogenic and anthropogenic VOCs in an oxidation flow reactor, collected SOA filter samples, and analyzed the samples with orbitrap mass spectrometry. SOA tracers identified in the samples were added to a mass spectral database. Ambient PM2.5 samples were likewise analyzed with the same orbitrap mass spectrometry technique and were screened for the SOA tracers identified in the laboratory OFR studies. About half of the signal of the CHO-containing compounds was attributed to those SOA tracers. A principal component analysis on the ambient samples was conducted that identified clusters of compounds corresponding to different atmospheric processes.

Overall this paper describes an innovative analytical approach that can aid in the interpretation of atmospheric oxidative aging processes and source apportionment. I support eventual publication of the paper after consideration of my comments below.

1. Can the authors explicitly comment on whether application of their aerosolomics data base is limited to PM2.5 samples that are analyzed specifically with a Thermo Fisher Orbitrap mass spectrometer and the related Compound Discovery software? I think that this is the case, but if I am wrong, it would be useful to clarify how/where else it can be applied.

2. **L75**. In addition to the irradiance, the external OH reactivity (OHR) also significantly influences the OH exposure (e.g. Li et al., 2015; Peng et al., 2015; Rowe et al., 2020). 50 ppb $SO_2$ was used in offline calibration experiments to constrain the OH exposure (Fig. S1). The corresponding OHR in the $SO_2$ calibration experiments is 50 ppb * 2.5e10 molec/cm3 * 9e-13 cm3/molec/s = 1 $s^{-1}$ . However, the OHR in the SOA studies was most likely considerably larger than 1 $s^{-1}$. Consequently, the OH exposure was probably lower than what was suggested from the calibration data due to OH suppression. For example, in the $\alpha$-pinene OFR experiment, ~83.9 $\mu g\ m^{-3}$ SOA was generated; assuming an SOA yield of approximately 0.3 (e.g. Lambe et al., 2015) and complete consumption of the $\alpha$-pinene, the initial $\alpha$-pinene concentration was approximately 83.9/0.3 = 280 $\mu g\ m^{-3} \approx 51$ ppb, with a corresponding OHR ~ 51*2.5e10*5.33e-11 = 68 $s^{-1}$. At these conditions, along with the experimental conditions provided in Section 2.1, I estimate that the corresponding OH exposure was approximately $1*10^{11}$ molec/$cm^{-3}$-sec using the OFR254 OH exposure estimation equation introduced by Peng et al. (2016) and published online here: https://sites.google.com/site/pamwiki/estimation-equations?authuser=0. This calculated OH exposure value is about 10 times lower than that obtained from their $SO_2$ calibration. This analysis should be applied to the other OFR experiments as well, and a column should be added to Table S1 with the corresponding calculated $OH_{exp}$ for comparison purposes. While the overall conclusions of the paper will remain unchanged, the (likely) lower photochemical age may provide higher confidence in applying the SOA tracers in the authors' aerosolomics database to ambient PM2.5 samples that may also be subject to lower aging timescales.

3. **L228**: The authors state: "Furthermore, ion source dimerization is a known phenomenon that hinders the unambiguous identification of atmospheric dimers, or leads to misinterpretation of results from direct-injection HESI." Can they provide references for this statement?

4. **L264**: The formulas that are listed for terpenylic acid and MBTCA are incorrect: terpenylic acid should be $C_8H_{12}O_4$, MBTCA should be $C_8H_{12}O_6$.

5. **L307** – **L316**: I find the discussion of the "unassigned" signals vague and unsatisfactory. It seems clear from the text that 53.5% of the signal in the ambient samples does not correspond to the SOA tracers obtained from the laboratory OFR studies. However, all that is stated is that "isoprene as well as other monoterpenes to be promising candidates closing this gap". Further, Figure 7 conflates "unassigned" with "unidentified", because information about compound MW, carbon number, and O/C is shown here, though it is difficult to interpret from the figure. At the least, they should list the formulas of the "few compounds with high signal intensities [that] remain unassigned" in the text, and perhaps add a table to the supplement with the corresponding information that is shown in Figure 7, rather than interpreting the result without providing the data to support the hypothesis as was done here.

**References**

R. Li, B.B. Palm, A.M. Ortega, J. Hlywiak, W. Hu, Z. Peng, D.A. Day, C. Knote, W.H. Brune, J. de Gouw, and J. L. Jimenez. Modeling the radical chemistry in an Oxidation Flow Reactor: radical formation and recycling, sensitivities, and OH exposure estimation equation. *Journal of Physical Chemistry A*, 119, 4418–4432, doi:10.1021/jp509534k, 2015.

Z. Peng, D.A. Day, H. Stark, R. Li, B.B. Palm, W.H. Brune, and J.L. Jimenez. HOx radical chemistry in oxidation flow reactors with low-pressure mercury lamps systematically examined by modeling. *Atmos. Meas. Tech.*, 8, 4863-4890, doi:10.5194/amt-8-4863-2015, 2015.

A. T. Lambe, P. S. Chhabra, T. B. Onasch, W. H. Brune, J. F. Hunter, J. H. Kroll, M. J. Cummings, J. F. Brogan, Y. Parmar, D. R. Worsnop, C. E. Kolb, and P. Davidovits. Effect of oxidant concentration, exposure time and seed particles on secondary organic aerosol chemical composition and yield. *Atmos. Chem. Phys.*,15, 3063-3075, 2015.

Rowe, J. P., Lambe, A. T., and Brune, W. H.: Technical Note: Effect of varying the $\lambda$ = 185 and 254 nm photon flux ratio on radical generation in oxidation flow reactors, Atmos. Chem. Phys., 20, 13417–13424, https://doi.org/10.5194/acp-20-13417-2020, 2020.

---

## Author Comment (AC1)

**Reply on anonymous Referee #2**

We thank referee #2 for the constructive comments that allowed us to improve the manuscript. Original comments are written in black, our replies in blue as well as comprehensible excerpts from the text including the old and the new version.

**General comments:**

It would be useful to say a few words about the historical origin of this approach in the introduction, as the atmospheric community may not be aware about this development.

A brief introduction to the topic of metabolomics has been included with the introduction.

> 50  Inspired by metabolomics, a tool widely used in the life sciences to identify metabolites, metabolic pathways, and biomarkers (Fiehn, 2002; Johnson et al., 2016), we created an *aerosolomics* database for database-assisted identification of marker compounds (without having the need for authentic standards), and hence enables the investigation of atmospheric transformation pathways of VOCs under different oxidation conditions. The database enables compound matching based on filters from potential aerosol mass (PAM) oxidation flow reactor (OFR) experiments of nine biogenic and anthropogenic VOCs. We applied the database to ambient air filter samples collected in summer 2018 near

**Specific comments:**

Line 162: Here, a reference to the paper in which MBTCA was first reported would be appropriate: Szmigielski R., et al., 2007.

The citation was added.

> 165 that this parameterization comprises a large molecular corridor and thus leads to a wide range of $\log_{10} C^*$. A bias has been reported for nitrogen containing compounds (Isaacman-VanWertz and Aumont, 2021), but also for CHO compounds it appears to be biased. For example, $\log_{10} C^*$ of the atmospheric tracer 3-methyl-1,2,3-butanetricarboxylic acid ($C_8H_{12}O_6$, MBTCA, Szmigielski et al. (2007)) results in $1.97 \, \mu g \, m^{-3}$, while with SIMPOL.1 (Pankow and Asher, 2008) we find $\log_{10} C^*$ at $298 \, K$

Line 179: The authors mention here that they were not able to determine the individual chemical structure of the different SOA compounds, but, in my opinion, it is feasible to assign most of them taking into account available knowledge, certainly for SOA related to alfa- and beta-pinene, on the basis of MS/MS and reversed-phase LC retention data. See below.

In general, we decided to use the identification scheme of Schymanski et al. (2014). Therefore, we speak only of unambiguously identified (level 1) compounds when we compared to authentic standards. In this case, this applies only for pinic acid, MBTCA, and phthalic acid. Certainly we are able to make level 2 identifications based on available knowledge from literature (MS/MS spectra for oxidation products of other terpenes and of accretion products). These level 1 and level 2 identifications help us to validate formation mechanisms of certain compound clusters of the hierarchical cluster analysis. For example MBTCA appears in our daytime OH-chemistry cluster. In our revised manuscript we include the majority of the suggested papers on single compound identification, although the aerosolomics approach does not rely on level 1 identifications of every single compound by authentic standards.

Lines 188 - 207: In this section, the oxidation products of alfa-pinene are discussed, which have been well documented in the literature. References to the early literature, in which these SOA products were first characterized on the molecular level, are appropriate. Many of the products mentioned have even been unambiguously assigned, not only tentatively. For example, the following molecular markers related to alfa- and beta-pinene:

Please consider the comment above.

$C_9H_{14}O_4$ (MW 186): characterized as pinic acid (e.g., Yu et al., 1999; Glasius et al., 2000)

The citation was added.

> in the mass range between 140 and 210 Da, dimers are in the range between 300 and 400 Da. The major products dur-
> 195 ing ozonolysis are (pinic acid ($C_9H_{14}O_4$ at 8.79 min), (, Yu et al. (1999), level 1), terpenylic acid ($C_8H_{12}O_4$ at 6.67 min),
> (, Claeys et al. (2009), level 2), pinyl-diaterpenyl ester ($C_{17}H_{26}O_8$ at 11.28 min, Kahnt et al. (2018), Yasmeen et al. (2010)
> , level 2), $C_8H_{14}O_5$ (at 5.84 min), and $C_8H_{14}O_6$ (at 6.56 min). Oxidation by OH reduces the absolute signal intensity of

$C_{10}H_{16}O_4$ (MW 184): characterized as hydroxypinonic acid (e.g., Glasius et al., 1999)

We mention $C_{10}H_{16}O_4$ (MW 200) in the 3-carene system which has a different retention time and MS/MS spectra than hydroxypinonic acid from alpha-pinene. According to Larsen et al. (2001) we added hydroxy-3-caronic acid and 3-caronic acid ($C_{10}H_{16}O_3$).

> Panel (d) shows the results of the 3-carene oxidation experiments. Three monomers are the most prominent products in both
> systems: (caric acid ($C_9H_{14}O_4$ at 9.62 min), (, Yasmeen et al. (2011), level 2), hydroxy-3-caric acid ($C_{10}H_{16}O_4$ at 8.72 min,
> 225 Larsen et al. (2001), level 3), and (3-caronic acid ($C_{10}H_{16}O_3$ at 10.27 min, Larsen et al. (2001), level 3). The four dimers

$C_8H_{12}O_6$ (MW 204): characterized as MBTCA (Szmigielski et al., 2007)

Please consider the comment "Line 162".

$C_8H_{12}O_4$ (MW 172): characterized as terpenylic acid (Claeys et al., 2009; Yasmeen et al., 2011)

The citation was added. Please consider the comment "$C_9H_{14}O_4$".

$C_8H_{12}O_5$ (MW 188): characterized as hydroxyterpenylic acid isomers (Kahnt et al., 2014)

The citation was added.

> formed, which is in general agreement with Hammes et al. (2019). The ozonolysis shows three major products, (hydroxyterpenylic acid ($C_8H_{12}O_5$ at 5.57 min), (, Kahnt et al. (2014), level 2), ketolimononic acid ($C_9H_{14}O_4$ at 6.44 min, Yasmeen et al. (2011) , level 2), and $C_{10}H_{16}O_5$ (at 6.85 min). In the OH system (at 6.44 min) ketolimononic acid becomes the major compound
> 220    whereas the intensity of (at 5.57 min) hydroxyterpenylic acid increases clearly. Analogous to the $\beta$-pinene oxidation, the $C_9H_{14}O_4$ isomer at 6.44 minutes ketolimononic acid can be used as specific limonene tracer due to the missing appearance of

$C_{17}H_{26}O_8$ (MW 358): characterized as cis-pinyl-diaterpenyl ester (Yasmeen et al., 2010; Kahnt et al., 2018)

The citations were added. Please consider the comment "$C_9H_{14}O_4$".

The same comment applies to molecular markers related to delta-3-carene and d-limonene, and to other molecular markers discussed in the text.

Designations and references have been added in some places where identification was possible, e.g., hydroxyterpenylic acid ,caric acid or ketolimononic acid.

Legend Figure 4: Molecular weight has no dimensions; delete "Da" in the x-axis.

Please consider the comment "Line 360"

Lines 279-283: Here, mention is made of CHNO biomass burning markers, such as nitrosalicylic acid and methylnitrophenol. It would be appropriate to cite the early papers in which products like methylnitrophenol were first reported in ambient fine aerosol by the group of Grgic, e.g., Kitanovski, Z., et al., J. Chrom. A 2012.

In order to limit our number of citation we decided to cite the first paper, and a more recent one.

> 290 despite proximity to an airport and a large refinery. Even though, several CHNO tracers for biomass burning could be identified, they only play a minor role with regard to noticeably lower signal intensities: $C_7H_5NO_5$ (nitrosalicylic acid, level 3), $C_7H_7NO_4$ (methylnitrocatechol, level 3), $C_7H_6N_2O_5$ (methyldinitrophenol/dinitrocresol, level 2), $C_7H_6N_2O_6$, $C_6H_5NO_4$ (nitrocatechol, level 2), $C_6H_4N_2O_5$ (dinitrophenol, level 2), $C_8H_7NO_4$ (methylnitrobenzoic acid, level 3), and $C_8H_9NO_5$ (Kitanovski et al., 2012; Salvador et al., 2021).

Line 344: The abundant $C_8H_{12}O_5$ product can be assigned to the alfa-pinene-related SOA markers, characterized as hydroxyterpenylic acid isomers. See Kahnt et al., 2014.

The citation has already been added, please consider the comment "$C_8H_{12}O_5$". At this point in the text we use the suggested compound name.

Line 360 and many places elsewhere: Molecular weight has no dimensions but molecular mass has; thus: ''The molecular mass reaches up to 440 Da and ….'' or "The molecular weight reaches up to 400 and …..".

Throughout the text, molecular mass with the unit Da has now been used uniformly.

Lines 371-374: Here, alfa-pinene-related dimers $C_{17}H_{26}O_8$ and $C_{19}H_{28}O_7$ are discussed; see the detailed chemical characterization study by Kahnt et al., 2018. It is also relevant to mention that the $C_{17}H_{26}O_8$ dimer, characterized as cis-pinyl-diaterpenyl ester, was first reported in nighttime ambient fine aerosol (Yasmeen et al., 2010).

The citation to Yasmeen et al. (2010) and their findings were added which support the correctness of the statistical approach. The citation to Kahnt et al. (2018) was added previously.

> 385 experiments described in Sect. 3.1. These 14 dimers are responsible for 72 % of the mean dimer signal intensity. Among them are pinyl-diaterpenyl ester $C_{17}H_{26}O_8$ (*aerosolomics*-database library: $\alpha$-pinene/$\beta$-pinene + $O_3$) and $C_{19}H_{28}O_7$ (*aerosolomics*-database library: $\alpha$-pinene/3-carene + $O_3$), both of which are described as esters between pinic acid and terpenylic acid (Gao et al., 2010) or pinic acid and hydroxypinonic acid (Müller et al., 2008), respectively. Furthermore, Yasmeen et al. (2010) reported pinyl-diaterpenyl ester in nightime ambient aerosol.

Line 395: Here, mention is made of a CHNOS compound with a mass of 295 Da. New intriguing insights about abundant MW 295 nitrooxy organosulfates related to alfa-pinene have more recently been gained by the group of Yu; see Wang, Y.-C., et al., Environ. Sci. Technol., 2021.

The citation was added.

> tion times indicates a broad mixture of polarities. Among the most common signals are  terpene-derived nitrooxy organosulfates like $C_{10}H_{17}NO_7S$ (Wang et al., 2021), isoprene-derived CHOS like $C_2H_4O_6S$ (Claeys and Maenhaut, 2021) as well as monoterpene oxidation products like MBTCA, terpenylic acid,
> 275  and pinic acid.

**Technical comments:**

Line 47: PM2.5 (2.5 in subscript)

Has been corrected.

> 45 isotopic pattern. Furthermore, $MS^2$-spectra can be compared to fragmentation libraries and enable database-assisted identification of compounds (Ditto et al., 2018; Ma et al., 2022; Pereira et al., 2021; Pleil et al., 2018). However, there are currently no established databases of atmospheric SOA tracers, which can be applied on measurements of ambient $PM_{2.5}$ filter samples.

Line 59 and many places elsewhere in the text: .... (98%, Alfa ....; remove the space before "%"

You are right, in English, no space is used, but the authors adhere to the AMT submission information (section "Mathematical notation and terminology"), which is based on the 9th edition of "The international System of Units (SI)" (2019). For this reason, a non-breaking space has been used.

Line 156: .... of this mode .....??

In this context a "node" is defined as a knot of a non-target analysis workflow.

Line 221: .... ß-norcaryophyllonic acid ??

Has been corrected.

In addition to the four monoterpenes, we investigated the composition of sesquiterpene-SOA from *trans*-caryophyllene

230   ($C_{15}H_{24}$). During ozonolysis we find one major and four minor products  in the mass range between 198 and 302 Da (Fig. 2e). The major compound is tentatively identified as $\beta$- norcaryophyllonic

acid ($C_{14}H_{22}O_4$ at 11.82 min, van Eijck et al. (2013), Jaoui et al. (2003), level 3). In

**Legend Fig. 3: .... The most abundant compounds .....**

Has been changed in the captions of Fig. 2 and Fig. 3 and in the text in line 191.

190   the ozonolysis products under dark conditions, while the lower spectra show the products from OH oxidation (254 nm UV).

 The most abundant compounds are labeled with the predicted formula and their

retention time, however, the database contains these entries of all compounds down to 1 % relative peak intensity.

**Figure 2.** Mass spectra of the detected products from the OFR experiments of five biogenic precursors (a) $\alpha$-pinene, (b) $\beta$-pinene, (c) limonene, (d) 3-carene, (e) and *trans*-caryophyllene. The intensity is normalized to the highest signal of each chemical system. The  most  abundant compounds of each experiment are labeled with their predicted composition and the according retention time.

**Figure 3.** Mass spectra of detected products from OFR experiments of four anthropogenic precursors (a) 1,2,4-trimethylbenzene, (b) toluene, (c) *o*-xylene, (d) and naphthalene. The intensity is normalized to the highest signal of each chemical system. The  most  abundant compounds of each experiment are labeled with their predicted composition and the according retention time.

**Line 350: However, it needs .....**

Has been changed.

It is also remarkable that compounds appearing in this cluster show low standardized values in sample cluster II. Low standardized values indicate lower signal intensities of the MS measurements. However, it  needs to be considered that signal

365   intensities are not directly quantitative to its concentration. It is well known, that the ionization efficiency of HESI varies

**References**

Larsen, B.R., Di Bella, D., Glasius, M. et al. Gas-Phase OH Oxidation of Monoterpenes: Gaseous and Particulate Products. Journal of Atmospheric Chemistry 38, 231–276 (2001). https://doi.org/10.1023/A:1006487530903.

---

## Author Comment (AC2)

**Reply on anonymous Referee #1**

We thank referee #1 for the constructive comments, which improved the manuscript especially with regard to the OH exposure estimates in the oxidation flow reactor. Original comments are written in black, our replies in blue as well as comprehensible excerpts from the text highlighting the tracked changes.

1. Can the authors explicitly comment on whether application of their aerosolomics data base is limited to PM$_{2.5}$ samples that are analyzed specifically with a Thermo Fisher Orbitrap mass spectrometer and the related Compound Discovery software? I think that this is the case, but if I am wrong, it would be useful to clarify how/where else it can be applied.

For individual substances, we see good comparability of fragmentation patterns with other instruments (e.g. QToF, Zhao et al. (2022) or linear ion trap LXQ, Yasmeen et al. 2010). Whether this comparability applies to the entire database and other mass spectrometers needs to be (and will be) investigated in future studies.

The application of the database, however, can be done with the related Compound Discoverer software, with the open source software MZmine 3 (https://mzmine.github.io/) as with every self-build program to match mass spectra. For this we provide the database as db-, msp-, and csv-files on our homepage. We added this information to the conclusion.

> 465    We would like to encourage the community to apply the database on their own samples. Therefore we provide the database as db-files, msp-files, and csv-files which allows the application of the database with Compound Discoverer, MZmine 3 or every self-build solution. As a community effort, further input to the database is desirable to improve our understanding of sources and formation of secondary organic aerosol.

2. L75. In addition to the irradiance, the external OH reactivity (OHR) also significantly influences the OH exposure (e.g. Li et al., 2015; Peng et al., 2015; Rowe et al., 2020). 50 ppb SO$_2$ was used in offline calibration experiments to constrain the OH exposure (Fig. S1). The corresponding OHR in the SO$_2$ calibration experiments is 50 ppb * 2.5e10 molec/cm3 * 9e-13 cm3/molec/s = 1 s$^{-1}$. However, the OHR in the SOA studies was most likely considerably larger than 1 s$^{-1}$. Consequently, the OH exposure was probably lower than what was suggested from the calibration data due to OH suppression. For example, in the $\alpha$-pinene OFR experiment, ~83.9 $\mu$g m$^{-3}$ SOA was generated; assuming an SOA yield of approximately 0.3 (e.g. Lambe et al., 2015) and complete consumption of the $\alpha$-pinene, the initial $\alpha$-pinene concentration was approximately 83.9/0.3 = 280 $\mu$g m$^{-3}$ ≈ 51

ppb, with a corresponding OHR ~ 51*2.5e10*5.33e-11 = 68 s$^{-1}$. At these conditions, along with the experimental conditions provided in Section 2.1, I estimate that the corresponding OH exposure was approximately 1*10$^{11}$ molec/cm$^{-3}$-sec using the OFR254 OH exposure estimation equation introduced by Peng et al. (2016) and published online here: https://sites.google.com/site/pamwiki/estimation-equations?authuser=0. This calculated OH exposure value is about 10 times lower than that obtained from their SO$_2$ calibration. This analysis should be applied to the other OFR experiments as well, and a column should be added to Table S1 with the corresponding calculated OHexp for comparison purposes. While the overall conclusions of the paper will remain unchanged, the (likely) lower photochemical age may provide higher confidence in applying the SOA tracers in the authors' aerosolomics database to ambient PM2.5 samples that may also be subject to lower aging timescales.

Thank you very much for this detailed comment. Based on your approach we calculated the external OH reactivity, the OH exposure as well as the equivalent atmospheric OH exposure. We changed the according section in the main part and in the SI.

was ~1 ppm, decreasing to 0.8 ppm under OH conditions.  Based on these experimental conditions we calculated the external OH  reactivity (Eq. S1 ) and the OH exposure using the OFR exposure estimator (Peng et al., 2015, 2016). The resulting OH exposures (Table S1) correspond to approximately 0.1–6 days of equivalent atmospheric OH exposure, based on the assumption of an averaged tropospheric OH concentration of $1.09 \times 10^6$ molecules cm$^{-3}$ (Li et al., 2018).

**Table S1.** Conditions during oxidation flow reactor experiments.

| Precursor | 254 nm lamp | Carrier gas | Source temperature | Mean mass concentration ± standard deviation | | OH exposure |
|---|---|---|---|---|---|---|
| | | | | Blank | Sample | |
| | V | ml min$^{-1}$ | °C | μg m$^{-3}$ | | molec cm$^{-3}$s$^{-1}$ |
| α-Pinene | 2 | 37.5 | 26 | 0.34 ± 0.09 | 83.9 ± 3.8 | $1.1 \times 10^{11}$ |
| α-Pinene | - | 37.5 | 26 | 0.34 ± 0.09 | 42.5 ± 1.6 | ~ |
| β-Pinene | 2 | 16.6 | 35 | 0.27 ± 0.08 | 184.4 ± 10.4 | $1.5 \times 10^{10}$ |
| β-Pinene | - | 16.6 | 39 | 0.27 ± 0.08 | 61.5 ± 8.9 | ~ |
| Limonene | 2 | 93.6 | 27 | 0.06 ± 0.07 | 104.3 ± 12.6 | $9 \times 10^{9}$ |
| Limonene | - | 93.6 | 27 | 0.06 ± 0.07 | 55.1 ± 3.2 | ~ |
| 3-Carene | 2 | 16.6 | 28 | 1.3 ± 0.4 | 62.5 ± 4.8 | $8.1 \times 10^{10}$ |
| 3-Carene | - | 12.9 | 29 | 1.3 ± 0.4 | 90.1 ± 10.6 | ~ |
| trans-Caryophyllene | 2 | 37.5 | 32 | 0.09 ± 0.05 | 52.5 ± 6.7 | $1.5 \times 10^{11}$ |
| trans-Caryophyllene | - | 71.6 | 32 | 0.09 ± 0.05 | 47.3 ± 4.8 | ~ |
| Toluene | 2 | 16.6 | 23 | 0.08 ± 0.03 | 66.2 ± 1.8 | $5.7 \times 10^{11}$ |
| o-Xylene | 2 | 25.4 | 22 | 0.42 ± 0.14 | 66.0 ± 2.2 | $3.2 \times 10^{11}$ |
| 1,2,4-Trimethylbenzene | 2 | 37.5 | 40 | 0.21 ± 0.08 | 24.2 ± 1.2 | $3.9 \times 10^{11}$ |
| Naphthalene | 2 | 93.6 | 25 | 2.9 ± 0.7 | 35.9 ± 5.7 | $4.1 \times 10^{11}$ |

We used the OFR exposure estimator (Peng et al., 2015, 2016) to approximate the OH exposure. Therefore, the OH reactivity ($OHR_i$) must be calculated (Eq. (S1)) using the mixing ratio $r_i$ (in ppb) and the rate constant $k_i$ from Atkinson and Arey (2003) of the respective precursor. The conversion factor $2.46 \times 10^{10}$ results from the ideal gas law (1 atm and 298.15 K).

$$OHR_i = r_i \cdot 2.46 \times 10^{10} \cdot k_i \qquad \text{(S1)}$$

5    The mixing ratio $r_i$ (in ppb) is calculated with Eq. (S2) from the mean mass concentration $c_i$ given in Table S1. The SOA yields ($y_i$) are estimated to be 0.3 for monoterpenes (Lambe et al., 2015), 0.6 for caryophyllene (Xavier et al., 2019), and 0.2 for aromatic precursors (Peng et al., 2022). In this approach, we assume complete consumption of the precursor. The molar volume is 24.47 L mol$^{-1}$ (1 atm and 298.15 K).

$$r_i = \frac{c_i}{y_i} \cdot \frac{\text{molar volume}}{\text{molecular weight}_i} \qquad \text{(S2)}$$

3. L228: The authors state: "Furthermore, ion source dimerization is a known phenomenon that hinders the unambiguous identification of atmospheric dimers, or leads to misinterpretation of results from direct-injection HESI." Can they provide references for this statement?

This phenomenon is actually visible in the raw data. As an example, Fig. S3 shows a high signal intensity of the dimer mass trace at the retention time of the monomer. This signal can be allocated to ion source dimerization.

4. L264: The formulas that are listed for terpenylic acid and MBTCA are incorrect: terpenylic acid should be $C_8H_{12}O_4$, MBTCA should be $C_8H_{12}O_6$.

Yes, the formulae were incorrect. However, due to changes based on the comments of the second referee no chemical formulas are used here anymore.

5. L307 – L316: I find the discussion of the "unassigned" signals vague and unsatisfactory. It seems clear from the text that 53.5% of the signal in the ambient samples does not correspond to the SOA tracers obtained from the laboratory OFR studies. However, all that is stated is that "isoprene as well as other monoterpenes to be promising candidates closing this gap". Further, Figure 7 conflates "unassigned" with "unidentified", because information about compound MW, carbon number, and O/C is shown here, though it is difficult to interpret from the figure. At the least, they should list the formulas of the "few compounds with high signal intensities [that] remain unassigned" in the text, and perhaps add a table to the supplement with the corresponding information that is shown in Figure 7, rather than interpreting the result without providing the data to support the hypothesis as was done here.

With "unidentified" we mean signals, for which we could not determine a molecular formula. "Unassigned" means that we determined a molecular formula, but the compound is not included in the database (we cannot assign a VOC precursor).

Figure 7 belongs to the results from the hierarchical cluster analysis and of these unidentified compounds only the molecular mass (not a molecular formula!) and retention time are available.

Figure S4 (= Fig. S5 in the old manuscript) shows all assigned and unassigned CHO compounds. Assigned means we have a match with the database. Unassigned means we can determine a CHO-formula, but we don't have a hit with the database. We have noticed that the identical coloring of unidentified (e.g., in Fig 7) and unassigned (in Fig. S5, old manuscript) can be confusing. For this reason, all unassigned CHO compounds in Fig. S4 (new manuscript) are white and hatched as in Fig. 5. Still we think that a discussion of the most abundant unassigned compounds is beneficial for interpretation and has been added to the text.

[Figure]

**Figure S4.** Assigned and unassigned CHO compounds from the representative selection of the field campaign samples. (a) Retention time vs. molecular mass. (b) Kroll diagram. (c) Van Krevelen diagram.

The majority of the assigned compounds from the representative selection of the Vienna field campaign have  molecular masses smaller than 250 Da and a retention time lower than 10 minutes (Fig. S4a). The mean number

320 of carbon atoms is 9 and the mean $\overline{OS}_C$ is $-0.4$ (Fig. S4b). SOA originated from biogenic precursors is located in the H/C area between 1.2 and 1.8 (Fig. S4c), while compounds with H/C < 1.2 can be of aromatic character. The observation that 19 % of the CHO compounds (number-wise) in the database are responsible for nearly 50 % of the mean signal intensity demonstrates the high relevance of the investigated VOCs in SOA formation. Nevertheless, a few compounds with high signal intensities remain unassigned, such as $C_8H_{14}O_5$, $C_7H_6O_2$, $C_9H_{16}O_4$ or the tentatively isoprene derived $C_4H_6O_5$ and $C_4H_8O_4$

325 (Claeys and Maenhaut, 2021; Krechmer et al., 2015). Considering the  retention behaviour and the molecular mass of these unassigned compounds (Fig. S4a) we expect isoprene as well as other monoterpenes to be promising candidates closing this gap. In addition, the precursors already used should be investigated under varying chemical conditions, like further oxidants or more complex mixtures of VOCs. A comprehensive study of isoprene oxidation is planned and the outcome will be uploaded to the *aerosolomics* database in the near future.

**References**

Zhao, Y., Yao, M., Wang, Y., Li, Z., Wang, S., Li, C., and Xiao, H.: Acylperoxy Radicals as Key Intermediates in the Formation of Dimeric Compounds in α-Pinene Secondary Organic Aerosol, Environ. Sci. Technol., Article ASAP, https://doi.org/10.1021/acs.est.2c02090, 2022.

Yasmeen, F., Szmigielski, R., Vermeylen, R., Gómez-González, Y., Surratt, J. D., Chan, A. W. H., Seinfeld, J. H., Maenhaut, W., and Claeys, M.: Mass spectrometric characterization of isomeric terpenoic acids from the oxidation of α-pinene, β-pinene, d-limonene, and Δ3-carene in fine forest aerosol, J. Mass Spectrom., 46, 425–442, https://doi.org/https://doi.org/10.1002/jms.1911, 2011.